# Targeting Epidermal Growth Factor Receptor (EGFR) in Pediatric Colorectal Cancer

**DOI:** 10.3390/cancers12020414

**Published:** 2020-02-11

**Authors:** Maria Debora De Pasquale, Alessandro Crocoli, Tamara Caldaro, Martina Rinelli, Gian Paolo Spinelli, Paola Francalanci, Raffaele Cozza, Alessandro Inserra, Evelina Miele

**Affiliations:** 1Department of Pediatric Onco-Hematology and Cell and Gene Therapy, IRCCS Bambino Gesù Children’s Hospital, 00165 Rome, Italy; mdebora.depasquale@opbg.net (M.D.D.P.); raffaele.cozza@opbg.net (R.C.); 2Surgical Oncology Unit, Department of Surgery, IRCCS Bambino Gesù Children’s Hospital, 00165 Rome, Italy; alessandro.crocoli@opbg.net (A.C.); alessandro.inserra@opbg.net (A.I.); 3Digestive Endoscopy and Surgery Unit, Department of Surgery, Bambino Gesù Children’s Hospital-IRCCS, 00165 Rome, Italy; tamara.caldaro@opbg.net; 4Department of Laboratories, Genetic Unit, IRCCS Bambino Gesù Children’s Hospital, 00165 Rome, Italy; martina.rinelli@opbg.net; 5Oncology Unit, Department of Medico-Surgical Sciences and Biotechnologies, Sapienza University of Rome, Via Giustiniano, 04011 Aprilia, Italy; gianpaolo.spinelli@uniroma1.it; 6Department of Laboratories, Pathology Unit, IRCCS Bambino Gesù Children’s Hospital, 00165 Rome, Italy; paola.francalanci@opbg.net

**Keywords:** anti-EGFR, panitumumab, childhood, colorectal cancer

## Abstract

*Background*: Colorectal carcinoma (CRC) is very rare in the pediatric and adolescent age range and clinical management is performed according to adult protocols. We report, for the first time in the literature, a case of a child with metastatic CRC successfully treated with panitumumab associated to chemotherapy. *Methods*: A twelve-year-old male was diagnosed with CRC with nodal metastasis and peritoneal neoplastic effusion. After performing a genetic evaluation, in light of the absence of mutations in *RAS* family genes, anti-Epidermal Growth Factor Receptor (EGFR) monoclonal antibody, panitumumab, was added to chemotherapy FOLFOXIRI. *Results*: The child successfully responded to therapy with normalization of the Carbohydrate Antigen (CA) 19.9 value after the third cycle of treatment. After the sixth cycle, he underwent surgery that consisted in sigmoid resection with complete D3 lymphadenectomy. At histological evaluation, no residual neoplastic cells were detectable in the surgical specimen. He completed 12 cycles of chemotherapy plus panitumomab and he is alive without disease 14 months from diagnosis. *Conclusions*: Our results suggest performing mutational screening for colorectal cancer also in the pediatric setting, in order to orient treatment that should include targeted therapies.

## 1. Introduction

Colorectal carcinoma (CRC) is the third most common cancer in Italian adults, representing 13% of all new diagnosis of cancer [1]. In children and adolescents, CRC is extremely rare, usually occurring as a part of hereditary syndromes, Lynch syndrome, familial adenomatous polyposis syndrome (FAP), juvenile polyposis (JP), Peutz-Jehgers syndrome (PJS), and Mutyh-associated polyposis (MAP) [2]. Over the past decades, adults affected by CRC have showed significant improvement in overall survival. This is in part due to the better understanding of the tumor’s molecular biology and the consequent use of molecular targets, and to more aggressive surgical approaches [3,4].

Clinical management of CRC in the pediatric population is performed according to adult protocols, though CRC in children and adolescents typically shows a worse prognosis mainly related to biological aggressiveness but also to late diagnosis and advanced stage at presentation [5,6,7].

We present a case of a twelve-year-old male with metastatic CRC successfully treated with chemotherapy associated to panitumumab, an anti-epidermal growth factor receptor (EGFR) agent. To our knowledge, this is the first report in the literature of a pediatric case treated with panitumumab. We widely reviewed the current literature about CRC in childhood and the tailored management of adult cancers occurring at a pediatric age.

Case: A twelve-year-old male was referred to our hospital emergency with repeated episodes of abdominal pain.

At medical history, he reported chronic constipation with occasional colic abdominal pain, two episodes of rectal bleeding two years and two months before, hyporexia, and weight loss of about two kilos (10% of his weight) in the last two months. No vomiting episodes were reported. Previous laboratory tests excluded food allergies and celiac disease. At physical examination, he was pale, cachectic, and suffering from pain in the right lumbar region. Abdominal ultrasonography showed right hydroureteronephrosis. A Computed Tomography (CT) scan confirmed hydroureteronephrosis with right ureter incorporated inside a retroperitoneal 14 × 7 × 4 cm unresectable nodal mass infiltrating multiple small bowel segments and a concentric thickening of the sigmoid walls, extended by about 6 cm, with the presence of centimetric calcification (Figure 1).

After an open biopsy of the nodal mass, the pathology revealed lymph-node metastasis from epithelial neoplasia, while the analysis of the peritoneal fluid, collected during surgery, revealed the presence of neoplastic cells. Colonoscopy revealed a large polypoid mass of the sigmoid colon, measuring about 5–7 cm in diameter and lying about 20 cm from the anal verge. The lesion almost completely occluded the colonic lumen, allowing a small endoscope (diameter 4.9 mm) to pass, albeit with some difficulty (Figure 2). Multiple biopsies were performed, with the pathology revealing undifferentiated adenocarcinoma (Figure 3). An evaluation of tumor markers showed a high level of carbohydrate antigen (CA) 19-9, 330 UI/mL (normal value < 37 UI/mL) while carcinoembryonic antigen (CEA) levels were normal.

According to the indication for adults with advanced colorectal adenocarcinoma, the child was started on chemotherapy with the FOLFOXIRI combination (Irinotecan 165 mg/mq over 1 h followed by oxaliplatin 85 mg/mq and leucovorin 200 mg/mq concomitantly over 2 h, on day 1, and followed by 5-fluorouracil 3200 mg/mq as a 48 h continuous infusion starting on day 1) every other week. At the same time, an evaluation of pan-Ras gene mutations was begun.

In detail, genetic studies were approved by the Bambino Gesù Children’s Hospital ethics committees. After obtaining informed consent for genetic testing, molecular characterization of the blood and tumor DNA of the patient was performed by next generation sequencing (NGS), using a custom panel (Roche NimbleGen, Madison, WI, USA), which comprises 339 targeted cancer-related genes. The BaseSpace pipeline (Illumina, https://basespace.illumina.com/) and TGex software (LifeMap Sciences, Inc., (Alameda, CA, USA) were used for the variant calling and annotating variants, respectively. DRAGEN Somatic Pipeline (Illumina, San Diego, CA, USA) was used for variant calling of low-allele frequency variant in tumor DNA. Variants identified as pathogenic were visualized by the Integrative Genome Viewer (IGV). Table 1 summarizes the prediction analysis and American College of Medical Genetics (ACMG) classification of the 13 driver oncogenes observed with the NGS analysis on the neoplastic tissue.

Sequence analysis of DNA extracted from the patient’s blood excluded the presence of germline variants in cancer predisposition genes, such as in Li-Fraumeni syndrome gene *TP53* (OMIM #151623), adenomas multiple colorectal gene *MUTYH* (OMIM #608456), adenomatous polyposis gene *APC* (OMIM #175100), and mismatch repair genes (MMR) *MSH2*, *MSH6*, *MLH1*, *MLH3*, and *PMS2*). Furthermore, high-coverage sequencing on the DNA extracted from the patient’s tumor sample did not reveal any detectable somatic variant in the therapeutic target genes *KRAS, NRAS*, and *BRAF* but showed the presence of a missense change in the *TP53* gene (NM_000546.5), c.733G>A (p.Gly245Ser), with an allele burden of 8% (according to NGS performed at high depth >500X) (Figure 4). This variant has been previously reported to segregate in multiple families with Li-Fraumeni syndrome (PMID: 1565143, 24122735, 17311302) and was found to be a somatic alteration in several types of cancer (COSM6932) [8]. Experimental studies have shown that this variant disrupts the ability of TP53 to bind to DNA and significantly decreases its transcriptional transactivation activity (PMID: 12826609, 20128691). This missense substitution can be classified as a pathogenic variant (class 5) according to the ACMG guidelines, on the basis of *PS3*, *PM1*, *PM2*, *PP3*, and *PP5* criteria [9].

After the first course, due to intestinal obstruction, the patient underwent right colostomy.

After two courses of FOLFOXIRI, in light of the absence of mutations in RAS family genes, the anti-EGFR monoclonal antibody, panitumumab, was added to chemotherapy at the dosage of 6 mg/kg intravenous over 1 h on day 1 of every course before starting FOLFOXIRI.

After the third course of chemotherapy (FOLFOXIRI plus panitumumab), the CA 19.9 level normalized.

A CT scan performed for the disease revaluation after six cycles of therapy showed more than a 75% reduction of the abdominal nodal mass and reduction of the concentric thickening of the sigmoid walls. A colonoscopy with multiple biopsies was performed. The pathology detected inflammatory infiltrated without residual neoplastic cells.

The child underwent surgery that consisted in sigmoid resection with complete D3 lymphadenectomy. At histological evaluation, no residual neoplastic cells were detectable in the surgical specimen.

After completing 12 courses, a complete re-evaluation of disease was performed, showing no evident residual disease, and colostomy closure was then performed.

Chemotherapy was well tolerated except for the appearance of nausea during drug infusion and for grade 2 mucositis (according to Common Terminology Criteria for Adverse Events (CTCAE) v5.0 Publish Date: 27 November 2017) about a week after each course. Nausea was controlled by administration of ondansetron, palonosetron before starting chemotherapy, and aprepitant. No vomiting episodes were recorded during chemotherapy administration under this combination of antiemetic drugs. Oral mucositis was treated with topical clorhydrate benzydamine and oral nistatine.

At the last follow-up (14 months from diagnosis), the child was alive in complete disease remission.

## 2. Discussion

Carcinoma of the large bowel is extremely rare in the pediatric age group [10]. It accounts for about 2% of all malignancies in patients aged 15 to 29 years [11]. Annually, in the United States, one case of CRC per one million persons younger than 20 years is reported and less than 100 cases are diagnosed in childhood [12]. Pediatric colorectal tumors can occur in any site in the large bowel. Ascending and descending colon tumors occur in approximately 30% of cases each, while rectal tumors are observed in approximately 25% of cases, as reported by larger case studies and reviews [7,13,14]. Abdominal pain is the most common symptom in children with descending colon tumors, followed by rectal bleeding, change in bowel habits, weight loss, nausea, and vomiting. Our patient experienced almost all these signs and symptoms. Right colon cancers can cause more treacherous symptoms but are usually associated with abdominal mass, weight loss, decreased appetite, blood in the stool, and iron deficiency anemia. The median duration of symptoms before diagnosis is reported in about 3 months, two months for our case [12,15]. The diagnostic workflow consists of clinical, laboratory, and radiographic studies. In detail, it should include the search for occult blood in the stool, evaluation of the liver and kidney function, tumor markers’ plasmatic levels (CEA, CA 19-9), and colonoscopy to detect neoplastic or pre-neoplastic lesions in the large bowel. Other ordinary imaging studies include barium enema or video capsule endoscopy followed by CT of the chest and bone scans [16].

Histologically, CRC of the pediatric and adolescent age (pCRC) shows a higher incidence of mucinous adenocarcinoma (40–50%), frequently with the signet ring cell type [10,12,15,17,18]. Tumors with such histology arise from the surface of the intestine, usually at the site of an adenomatous polyp. The tumor can extend into the intestinal muscle layer, or it can completely perforate the bowels, thus disseminating in the peritoneal cavity, or metastasize to the lymph nodes, liver, and ovaries in females [19,20]. These features of biological aggressiveness together with less responsiveness to chemotherapy may explain the worse prognosis typically shown by pCRC. Indeed, in the adolescent and young adult population with the mucinous histology, there is a higher incidence of signet ring cells, microsatellite instability, and mutations in the mismatch repair genes [18,21,22].

Furthermore, diagnosis is often late and young patients usually show an advanced stage at presentation [15], either as a gross tumor or as microscopic dissemination in the lymph nodes or on intra peritoneal organs [17,23]. Indeed, age younger than 21 years is considered as significant predictor of increased mortality [18]. Survival is consistent with the advanced stage of disease observed in most children with colorectal cancer, with an overall mortality rate of approximately 70%. For patients with a complete surgical resection or for those with low-stage/localized disease, survival is significantly prolonged, with the potential for cure [13].

pCRC often occurs as a part of hereditary syndromes [2,24]. The well-described forms of hereditary CRC are: 1) Polyposis (including familial adenomatous polyposis [FAP] and attenuated FAP, caused by pathogenic variants in the *APC* gene; and MUTYH-associated polyposis, caused by pathogenic variants in the *MUTYH* gene (MAP); and 2) hereditary nonpolyposis colorectal cancer or Lynch syndrome, caused by germline pathogenic variants in DNA mismatch repair genes (*MLH1*, *MSH2*, *MSH6*, and *PMS2*) and *EPCAM* [25,26]. Other colorectal cancer syndromes and their associated genes include Peutz–Jeghers syndrome (PJP) (*STK11*) [25], juvenile polyposis syndrome (JP) (*BMPR1A*, *SMAD4*), oligopolyposis (*POLE*, *POLD1*) [26], *NTHL1* [27], and Cowden syndrome (*PTEN*).

Of note, pCRC in the context of cancer predisposition syndromes has been shown to be associated with excellent outcomes [28].

In our patient, we excluded the presence of germline variants in cancer predisposition genes *MUTYH*, *APC,* and mismatch repair genes (*MMR*).

CRC in children and adolescent with non-inherited sporadic tumors usually lack *KRAS* mutations and other cytogenetic anomalies observed in adulthood [29]. Also, in our patient, we did not detect mutations in *RAS* family genes, but the presence of a missense pathogenic change in the *TP53* gene, probably an expression of the advanced disease stage.

Clinical management of CRC in the pediatric population is performed according to adult protocols and includes surgery, radiotherapy, chemotherapy, and targeted therapy. Complete surgical excision is the uppermost prognostic factor, but it could not be achieved in most instances. Tumor debulking can provide benefits for patients with widely metastatic disease [12]. In adults, current medical therapy includes the use of radiation for rectal tumors, together with 5-fluorouracil (5-FU)-based chemotherapy [30]. 5-FU-based regimen are at the mainstream of colon cancer therapy, usually associated with other agents, including irinotecan, oxaliplatin, and the anti-angiogenic agents bevacizumab or aflibercept [31,32,33]. Recently, monotherapy with regorafenib (an *RAF* inhibitor) has been shown to improve progression-free survival in patients with previously treated metastatic colorectal cancer. No significant benefit has been determined for interferon-alfa given in conjunction with 5-FU/leucovorin [34]. Other active agents used in adults include anti-EGFR monoclonal antibodies cetuximab [35] and panitumumab. The latter is a fully human recombinant monoclonal Immunoglobulin 2 (IgG2) antibody directed against the epidermal growth factor receptor, which has been principally approved for the treatment of patients with wild-type *KRAS* (exon 2 in codons 12 or 13) metastatic colorectal cancer [36]. Of note, right-sided colorectal cancers have been reported to not benefit from anti-EGFR therapies in the first-line metastatic setting. A possible explanation lies in the different embryogenesis of left-sided tumors [30]. Currently, in first-line settings, therapy for left-sided *RAS* and *RAF*-wild-type metastatic colorectal cancer can exploit either anti-EGFR agents or anti-Vascular Endothelial Growth Factor (VEGF) agents (bevacizumab) [30]. The sequence of use of biological drugs is controversial. In practice, anti-EGFR agents might be the favorite for left-sided *RAS* and *RAF*-wild-type tumors, mostly for patients, in whom response could lead to surgery [30,37].

Our patient was affected by a left-sided, locally advanced, pan*RAS/RAF* wild-type colorectal tumor. All these features, in light of the literature evidence in the adult population, pushed us to direct the treatment choice toward an aggressive chemotherapy regimen and to include the anti-EGFR panitumumab in the therapy plan. The final aim was indeed to lead the patient to surgery and to improve survival. We obtained a complete pathological response with acceptable toxicities.

Notably, a review of nine clinical trials including 138 patients aged less than 40 years showed that the use of combination chemotherapy improved progression-free survival and overall survival in this class of patients. Furthermore, overall survival and response rates to chemotherapy were comparable to those observed in older patients [38].

The genomic landscape of pediatric cancer is becoming increasingly well-defined and there is now an extensive list of recurrent genetic alterations with potential diagnostic, prognostic, or predictive value [39]. Sequential testing of single genes using standard methods could be unfeasible due to lacking available material and high costs. NGS offers a solution to this problem.

In our pediatric institution, we built up a rapid workflow, assessing by NGS most of the currently targetable molecules and the most frequent cancer predisposition genes that can be broadly applied to all pediatric tumors. Such sequencing strategies are becoming easily workable and extremely useful, in order to support innovative clinical trials in high-risk patients with unmet need [39]. Indeed, sequencing is not invasive if sample is available, tissues are already biopsied for pathological disease diagnosis, and blood sample can be easily obtained, too.

An example of success in this sense is represented by the screening and druggability of neurotrophic tyrosine receptor kinase (NTRK) gene fusions. These are oncogenic drivers induced by genomic rearrangements between the kinase domain of one of three tropomyosin receptor kinases (TRK) and a dimerization domain by another gene. The resulting fusion proteins are targetable with TRK inhibitors whose efficacy has also been proved on many childhood patients [40].

Our experience supports the usefulness of performing mutational screening for pediatric tumors and in particular for the rarest ones, such as colorectal cancers, in order to orient treatment that should include targeted therapies.

## 3. Conclusions

Colorectal carcinoma is an exceptional event in childhood and adolescence. Such rarity does not allow standardized therapy protocols and clinical management is based on evidence in adult patients. To our knowledge, this is the first report of a child with metastatic CRC successfully treated with panitumumab associated to chemotherapy. FOLFIRI-panitumumab was a well-tolerated regimen, which led to surgical resectability and complete pathological response in left-sided panRAS/RAF wild-type pediatric colorectal tumor. Mutational screening and targeted therapies must orient treatment for colorectal cancer also in the pediatric setting.

## Figures and Tables

**Figure 1 cancers-12-00414-f001:**
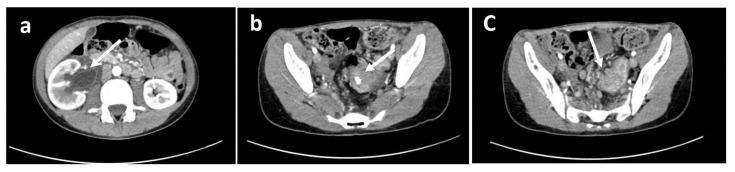
Computed Tomography images showing (**a**) right globular kidney site of hydro-ureteronephrosis; the ureter appeared markedly dilated up to the iliac tract where it was incorporated by a solid tissue with post-contrast enhancement of likely lymph node nature. This tissue extended from the right renal pelvis into the peri-ureteral adipose tissue (overall extension of about 14 × 7 × 4 cm). (**b**) a concentric thickening of the walls of the sigmoid was also evident, extended for about 6 cm, with the presence of centimetric calcification in the context. (**c**) multiple lymph nodes with rounded morphology, partially confluent were present in the perisigmoid meso with a maximum size of about 16 mm.

**Figure 2 cancers-12-00414-f002:**
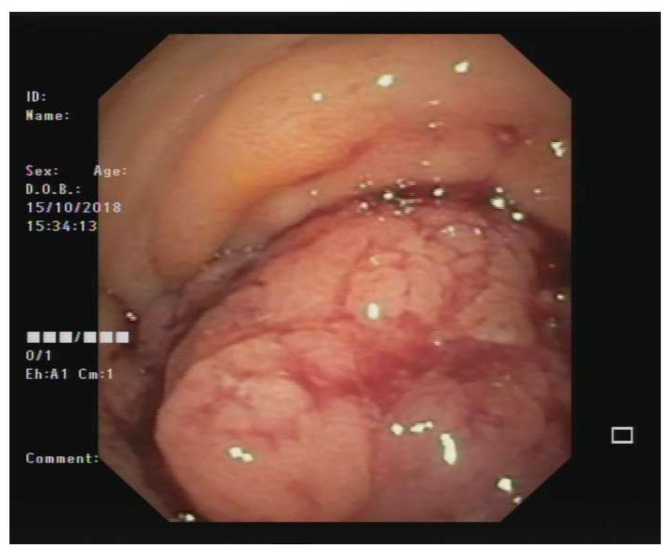
Colonoscopy image showing the large polypoid mass of the sigmoid colon (about 20 cm from the anal verge) that almost completely occluded the colonic lumen.

**Figure 3 cancers-12-00414-f003:**
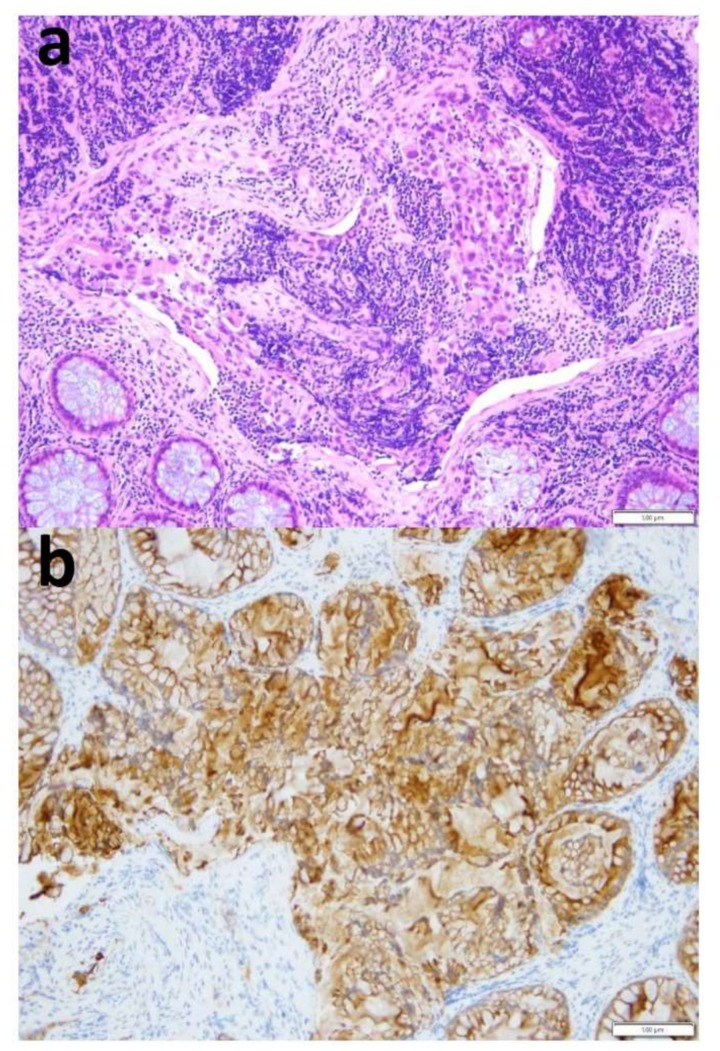
Histology. (**a**) Hematoxylin and eosin staining at 20× magnification showing fragments of colic mucosa in which there are foci of cells with atypical, polymorphic and polymetric nuclei, hyperchromic and sometimes nucleolated, coherent, with poorly differentiated carcinoma- (**b**) Immunohistochemical staining for CK CAM5.2 of the neoplastic cells at 20× magnification. Scale bar 200 µm.

**Figure 4 cancers-12-00414-f004:**
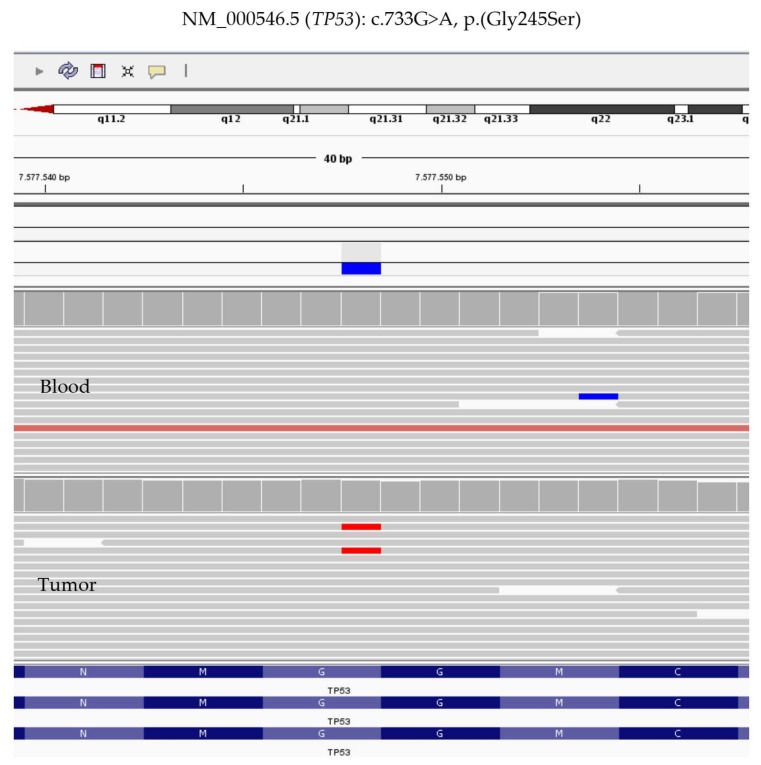
Integrative Genome Viewer (IGV) screen with the variant identified in blood (upper) and tumor DNA (down). Note the allele burden greater than 8% supporting the mosaic alteration of *TP53* variant. Scale bar: 100 µm.

**Table 1 cancers-12-00414-t001:** Prediction analysis and American College of Medical Genetics (ACMG) classification of the variants in 13 driver oncogenes, observed with next generation sequencing (NGS) analysis on the tumor tissue.

Genomic and Genetic DATA
GENE	REFSEQ	HGVS	DBSNP	ACMG	Location
EPCAM	NM_002354.2	c.-40C > T	rs747979626	VUS	2:47596605
MSH6	NM_000179.2	c.116G > Ap.Gly39Glu	rs1042821	BEN	2:48010488
TGFBR2	NM_001024847.2	c.118G > Ap.Asp40Asn	rs61732532	VUS	3:30664714
TGFBR2	NM_001024847.2	c.455-4T > A	rs11466512	BEN	3:30713126
MSH3	NM_002439.4	c.154_171del p.Ala52_Ala57del	rs201874762	BEN	5:79950699
MSH3	NM_002439.4	c.196_204del p.Pro66_Ala68del	rs879531814	BEN	5:79950741
MSH3	NM_002439.4	c.359-7G > A	rs1382543	BEN	5:79960955
APC	NM_000038.5	c.5465T > Ap.Val1822Asp	rs459552	BEN	5:112176756
PMS2	NM_000535.6	c.2007-4G > A	rs1805326	BEN	7:6022626
EGFR	NM_005228.3	c.1562G > Ap.Arg521Lys	rs2227983	BEN	7:55229255
MLH3	NM_014381.2	c.2531C > Tp.Pro844Leu	rs175080	BEN	14:75513828
MLH3	NM_014381.2	c.1870G > Cp.Glu624Gln	rs28756986	BEN	14:75514489
NRAS	NM_002524	no variants found			
KRAS	NM_033360	no variants found			
BRAF	NM_004333	no variants found			
APC	NM_000038	no variants found			
MLH1	NM_000249	no variants found

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
