# Peer review of "Targeting Epidermal Growth Factor Receptor (EGFR) in Pediatric Colorectal Cancer"

_cancers, 2020, doi:10.3390/cancers12020414_

Round 1

Reviewer 1 Report

This manuscript is case report about epidermal growth factor receptor (EGFR) targeting therapy in pediatric colorectal cancer patient. EGFR targeted therapy with chemotherapy FOLFOXIRI is nothing special about colorectal cancer therapy in adult, but this is extremely rare case in pediatric colorectal patient. This manuscript has important suggestions about anticancer therapy guideline for pediatric colorectal cancer patients. I have some minor comments for Authors

Figure 1 need to arrow indication on CT image for understanding tumor mass with high resolution CT image also need in this manuscript. Authors have to describe more detailed treatment schedule after 1st diagnosis. More detailed treatment history will be very helpful to understand about this rare case. Authors have to change Figure 3 to more high resolution image. And summarized information (figure or table) about genetic analysis data also need in this manuscript to understand this case. Amplification may be important with mutation characters in this case about several driver oncogenes to compare previous cancer cases. Toxicity profiling is very interesting issue in this case according to panitumumab and FOLFOXIRI combination therapy in pedicatric colorectal cancer. About adverse effects, more detailed description could suggest important guideline for further cases.

Author Response

We thank the reviewer for his/her careful reading of the manuscript and his/her constructive remarks. We have taken the comments on board to improve and clarify the manuscript. Please find below a detailed point-by-point response to all comments (reviewer’s comments in black, our replies in blue).

Reviewer 1

Comments and Suggestions for Authors

This manuscript is case report about epidermal growth factor receptor (EGFR) targeting therapy in pediatric colorectal cancer patient. EGFR targeted therapy with chemotherapy FOLFOXIRI is nothing special about colorectal cancer therapy in adult, but this is extremely rare case in pediatric colorectal patient. This manuscript has important suggestions about anticancer therapy guideline for pediatric colorectal cancer patients. I have some minor comments for Authors

Authors’ reply: we thank the reviewer for appreciating our manuscript.

Figure 1 need to arrow indication on CT image for understanding tumor mass with high resolution CT image also need in this manuscript.

Authors’ reply: High resolution CT images have been uploaded and arrows indicating tumor mass, lymph nodes and hydro-ureteronephrosis have been added.

Authors have to describe more detailed treatment schedule after 1st diagnosis. More detailed treatment history will be very helpful to understand about this rare case.

Authors’ reply: As suggested, the clinical history and treatment schedule have been detailed (see page 2, lines 54-58 and 74-78; page 3, lines 105-106)

Authors have to change Figure 3 to more high resolution image.

Authors’ reply: High-resolution image has been uploaded for ex-Figure 3, now Figure 4.

 And summarized information (figure or table) about genetic analysis data also need in this manuscript to understand this case. Amplification may be important with mutation characters in this case about several driver oncogenes to compare previous cancer cases.

Authors’ reply: As suggested, Table 1 summarizing genetic alterations analyzed and found in tumor tissue has been included (see new Table 1 and page 3 lines 87-88 in the main text)

Toxicity profiling is very interesting issue in this case according to panitumumab and FOLFOXIRI combination therapy in pediatric colorectal cancer. About adverse effects, more detailed description could suggest important guideline for further cases.

Authors’ reply: according to reviewer suggestions, the toxicity profile of FOLFOXIRI+Panitumumab has been detailed (see page 3, lines 120-123 of the main text)

Reviewer 2 Report

This is a well written article. I have a few comments.

Histologically it showed that the tumor was undifferentiated carcinoma. Was is confirmed by IHC hat it was indeed a poorly differentiated adenocarcinoma. Histology pictures with IHC should be added to the report. The report clearly identified the use of anti EGFR therapy along with chemotherapy for the treatment and that has been very well explained. This treatment protocol can be helpful for such population who don't have any genetic predisposition which again is well documented. Discussion can be small as we are only discussing the general population and very rarely comparing it with the case.

Author Response

We thank the reviewer for his/her careful reading of the manuscript and his/her constructive remarks. We have taken the comments on board to improve and clarify the manuscript. Please find below a detailed point-by-point response to all comments (reviewer’s comments in black, our replies in blue).

Reviewer 2

Comments and Suggestions for Authors

This is a well written article. I have a few comments.

Histologically it showed that the tumor was undifferentiated carcinoma. Was is confirmed by IHC hat it was indeed a poorly differentiated adenocarcinoma. Histology pictures with IHC should be added to the report.

Authors’ reply: We thank the reviewer for his/her positive comments on out manuscript and his/her constructive remark. As suggested, histology pictures with IHC have been added to the report (see new Figure 3)

The report clearly identified the use of anti EGFR therapy along with chemotherapy for the treatment and that has been very well explained. This treatment protocol can be helpful for such population who don't have any genetic predisposition which again is well documented.

Authors’ reply: we thank the reviewer for underlining this important issue.

Discussion can be small as we are only discussing the general population and very rarely comparing it with the case.

Authors’ reply: as suggested some parts of the discussion section have been shortened.

Reviewer 3 Report

The authors report a child CRC case who was successfully treated with panitumumab plus FOLFOXIRI based on molecular profiling.  This study suggests that the precision oncology approach is feasible even in childhood and shows a typical case with a good treatment course. However, this manuscript has several problems to be addressed.  

This manuscript looks like a case report. Manuscript type for this journal may be incorrect. The authors conclude that mutational screening is feasible even in the pediatric setting. However, all clinicians may already take this issue for granted because sequencing is not invasive if sample is available (in many case, tissues are already biopsied for pathological disease diagnosis, also blood sample can be easily obtained). In fact, clinical studies showing the efficacy of trk-inhibitors include many childhood patients and these drugs have already been administered to children all over the world. Thus, this conclusion is not novel or relevant. Eventually this patient was treated with FOLFOXIRI plus panitumumab, a standard regimen for adults, based on the absent of ras mutations. But, immunohistochemistry can also evaluate ras gene mutation status. Please clarify any additional merit of molecular testing (by using NGS) on this patient if the authors claim the importance of mutational screening or targeted therapy.

Author Response

We thank the reviewer for his/her careful reading of the manuscript and his/her constructive remarks. We have taken the comments on board to improve and clarify the manuscript. Please find below a detailed point-by-point response to all comments (reviewers’ comments in black, our replies in blue).

Reviewer 3

Comments and Suggestions for Authors

The authors report a child CRC case who was successfully treated with panitumumab plus FOLFOXIRI based on molecular profiling.  This study suggests that the precision oncology approach is feasible even in childhood and shows a typical case with a good treatment course. However, this manuscript has several problems to be addressed.  

Authors’ reply: We would like to thank the reviewer for his/her thoughtful comments and efforts towards improving our manuscript.

This manuscript looks like a case report. Manuscript type for this journal may be incorrect.

Authors’ reply: we took advantage from this case report to review the available literature on the field and to comment on the usefulness of target sequencing in colorectal cancer in childhood and in pediatric tumors more in general.

The authors conclude that mutational screening is feasible even in the pediatric setting. However, all clinicians may already take this issue for granted because sequencing is not invasive if sample is available (in many case, tissues are already biopsied for pathological disease diagnosis, also blood sample can be easily obtained). In fact, clinical studies showing the efficacy of trk-inhibitors include many childhood patients and these drugs have already been administered to children all over the world. Thus, this conclusion is not novel or relevant.

Authors’ reply: we thank the reviewer for this hint of discussion. We have commented on this issue (see pages 5; lines 207-210 and 217-221)

Eventually this patient was treated with FOLFOXIRI plus panitumumab, a standard regimen for adults, based on the absent of ras mutations. But, immunohistochemistry can also evaluate ras gene mutation status.

Please clarify any additional merit of molecular testing (by using NGS) on this patient if the authors claim the importance of mutational screening or targeted therapy.

Authors’ reply: Being our institution a pediatric center it usually faces only pediatric diseases. Sometimes it is hard to evaluate immunohistochemical markers that are specific for the adult population. For example, we do not have easily access to immunohistochemistry for RAS mutations. On the other side, we built up a rapid work-flow assessing by NGS all currently targetable molecules and the most frequent cancer predisposition genes that can be broadly applied to all pediatric tumors (see new Table 1 and comment on page 5, lines 211-216 ). We believe that this model could be exported and applied in other clinical realities. We commented also on this raised point.

Round 2

Reviewer 2 Report

Looks good. All question answered

Reviewer 3 Report

It looks the manuscript has been improved by the reviewers' comments.